# Hannah Arendt's Action Theory, Aesthetics and Feminist Curatorial Praxis

**Neda Mohamadi**

Art and Design Department, Middlesex University, London NW4 4BT, UK; nm1287@live.mdx.ac.uk

**Abstract:** This article considers the relationship between action (Arendt) and aesthetics in curatorial projects with feminist concepts. I suggest that Hannah Arendt's theory of action provides the connection between aesthetics and the notion of action in feminist curatorial praxis. The vision of feminism discussed here refers to the desire to understand matters from the specific point of view of women and considers the distribution of power and potentiality in various levels of life. The feminist theory in this research aims to reveal, show, and transform cultural, historical and social structures. From a broader perspective, living in the neoliberal realities alongside capitalist and patriarchal state structures provides multiple reasons and a rationale for collectively forming a new foundation of resistance. Feminism emerges in and through curatorial actions involving varied artistic expressions of freedom, discontent, etc. Four case studies concerning women as subjects are investigated, whose subject is migration and border-crossing, and both works and exhibitions are compared in terms of their curatorial approach, the level of action and their aesthetics methods.

**Keywords:** Hannah Arendt; aesthetics; feminist curation





## 1. Introduction

What is the relation between aesthetics and the "feminist turn in curatorial praxis"? The latter term that has been used in a number of academic research, exhibitions, conferences and publications (Fisher 2006; Deepwell 2006; Krasny 2015). How can Hannah Arendt's action theory be used to advance or describe feminist curatorial concerns? Finally, how has this debate been transferred into practice in curating arts? How does any notion of "the curatorial" link to feminist thought? For Martinon, the "curatorial is a jailbreak from pre-existing frames, [ . . . ] a strategy for inventing new points of departure, a practice of creating allegiances against social ills, a way of caring for humanity, the process of renewing one's own subjectivity". (Martinon 2015, p. 4). My understanding of "the curatorial" has roots in a philosophical contemplation and behavioral praxis on art about social and political subjects, and here I am concerned with how feminist theory converts itself into action with the hope of making a change in society and going beyond the mere pictorial representation of what is wrong.[1] Accordingly, the projects discussed are wide-ranging by both artists and curators and take the form of writing, visual artworks, interactive or noninteractive performances, talks, research projects or projected models for a future society.

There are very few studies of any depth on the potential benefits of Hannah Arendt's theories in arts and curatorial activities (Etzold 2011; Birchall and Sack 2014; Holert 2020) but even less in productions where feminist subjects are the main focus of art (Feldman 2015). I have chosen a deductive approach that moves from Arendt's theory toward the practice to re-evaluate the implications of this theory in the selected case studies for art and curation of art.

## 2. The Feminist Turn in Curatorial Praxis

In the neoliberal world we presently inhabit, "the curatorial" is a term that unites certain imaginary productions. Several scholars, such as Dorothee Richter, put their efforts

into defining curating as a process that lights the way between thinking as a process and the intention to make a change in curatorial and feminist theory and the world. Richter suggests "Curating means that the most diverse artefacts, installations, objects, events, performances, screenings and texts are combined and introduced into a new constellation. From that perspective, curatorial work can be described as the experiment of juxtaposing art with up-to-date concerns to improve or change the existing situation" (cited in Kolb 2019). In *The Curatorial: A Philosophy of curating*, Jean-Paul Martinon looks at curating's social function and indicates that the curatorial is an act of sending-off since it pushes curatorial action out of its comfort zone and this is why it never solely belongs to an institution, the dominant power or those who already enjoy the various privileges in society, whether an elite or the bourgeoisie (Martinon 2015).

Accordingly, two crucial points are emphasised here. First is the use of curatorial praxis dedicated to subjects which target a broader subject matter that influences "Other People"; and second, the nature of the praxis, which although existing in a present time has its face fixed toward the future. This approach toward the future in curating praxis is more discursive and radical than simply an expression of utopian desires. Equally, the potential for dialogue, exchanging ideas, and movements between different times, sites, and perspectives as provided by feminist theory is central to theorising, and exercising feminist approaches to the curatorial. Curatorial praxis can become activated in the social and cultural gaps between existing models of knowledge and because it is linked to the domain of thinking, it is able to create a space for a new language in arts to examine theories in real life.

Feminism is grounded in "the conviction that gender has been, and continues to be, a fundamental category for the organization of culture. Moreover, the pattern of that organization usually favors men over women" (Reckitt and Phelan 2001). Historically, feminist protest is identified with women creating a situation in which they can get access to their rights and have their voices heard. As Rosemarie Tong suggests, feminism is a coalition of different political groups: "They signal to the public that feminism is not a monolithic ideology and that all feminists do not think alike" (Tong 2009). Where feminist curatorial projects try to change and expand established knowledge and definitions, it is important to recognize how they present and update concepts based on this progressive knowledge with an eye on the future and the foundation they create for days to come (Richter 2008).

This article suggests a link between curatorial praxis and politics in feminism that can also ''push open the concept of what counts as political by looking at a wide range of actions and activities as well as the varied places and spaces" (Tong 2009) where these actions can be performed. Hence, feminism's turn in curatorial praxis is attempting through trial, error and evaluation at every single step to highlight the situation of women and notions of sexual difference as well as topics in feminist politics. It gives up the safety of what is known and accepts a transformational movement from historically fixed patriarchal notions towards change.

## 3. Arendt's Action Theory and Feminists' Tendency in Curatorial Praxis

The implicit reason for pursuing positive and negative approaches to Arendt's school of thought is this philosopher's ground-breaking theories in her method of choosing and tackling the subject of the action in politics and, in the next step, its relationship with aesthetics. Hannah Arendt is a philosopher of gaps. She addressed many complex problems directly and this point links her to many contemporary debates in political and critical theory that can be used in the discussions on curatorial action, aesthetics, and most of all feminist desires to make changes in society. However, Hannah Arendt as a philosopher does not have a good reputation among feminists since, in most of her works, she is not only a gender-neutral philosopher but also has been seen as an anti-feminist thinker (Maslin 2013). This article does not try to prove the opposite statement—that she was a feminist; however, some of her theories can be considered a practical approach toward many subjects.

The current argument focuses on one of those crossing points that can combine ideas for the sake of studying curation of migration and art.

Action can be seen as a term for a broad distinction that applies to the things that merely "happen" to people—the events they undergo—and the various things they genuinely "do". The latter event, the doing, is the "act" or "action" (Wilson and Shpall 2016). Arendt determines action is related to three categories in "vita active" (active life): namely, labour, work, and action. Labour is the activity linked to the human condition in everyday life. Work is the activity that is related to the condition of worldliness. Finally, action specifies how these activities are linked to groups of people operating differently in and through specific forms of action (Arendt 1969). This claim is in line with Susan Archer Mann's definition of feminist thought that highlights "the social agency involved in theory production—how constructing theory is a social practice and a form of labor" (Archer Mann 2012).

Archer Mann's perspective toward the invention of social agency for making what can be called the social constitutions for publics is aligned with Arendt's theory. Consequently, the act of thinking directed towards the aim of social practice is an action and, at the same time, a constant process. For Arendt, action is also explicitly linked to the public's desire, either by thinking together, collective arguments or doing something in favour of the public. An action has no recognition as a singular or generalizable type of activity, nor can it be identified by who performs it in isolation from other members of society. Arendt implies action as the only type of activity that goes on directly between humankind without the intermediary of things or matter. It corresponds to the human condition of plurality, as well as to the fact that humans can act as a collective community, and not just as individuals who live on the earth and inhabit the world (Arendt 1969). Accordingly, the two central notions of action are freedom and plurality. Action can only be understood in the communal version of society and by grasping and analysing this notion of action, any concept of action by one individual becomes impossible. In this regard, action can be recognized as similar to language in that both become activated only in a collective community.

Action is always linked to power and power needs to be understood as "a set of actions that influences the actions of others" (Garnar 2006). Power controls the space in which action wants to function and actions are contingent on the realm of power. Power's contingency needs to set action down and let it go, and it is here that it links to curation, as indicated by Sarah Pierce, who suggests that the "setting down" of an artwork is the responsibility of the producer—artist, curator, etc.—and the next stage is "to go out", which happens either in the space of the art project or its presentation and is an action related to the public who are influential in the era of a project (Pierce 2015, p. 102).

Among Arendt's commentators, only a few scholars and curators have addressed her theory within this field and found similarities between the main features of curatorial praxis and action definition from Arendt (Etzold 2011; Feldman 2015; Holert 2020). Etzold, for example, focused on an idea from Jean-Luc Nancy, who describes the irrepresentability of community and defines practice or action as a notion that is directly linked to the *bios* and life; however, for Arendt, action belongs only to the collective and consequently is related to the political act and plural version of the community (Etzold 2011). Here, the matter is how and when artists create complex projects that require a curator to follow their lead and how and when curators bring artists together to create artistic ensembles of meaning. This is why the curatorial does not refer to being a member of a profession, but to a way of artistic thinking and behaving with the aim of producing/representing collective action.

Arendt's action theory relies on the present but it is also a process that directly influences the future. Unlike other philosophers, Arendt's philosophy has rarely been central to curators' work and so few directly refer to her philosophies since her philosophy considered more relevant to the social and political science than art (Birchall and Sack 2014). However, some art projects do feature and make use of challenging concepts from Arendt's

theories, such as human rights and marginalization; these topics are pretty common, and some relevant examples have been used in the text as case studies.

Hito Steyerl, in *The Wretched of the Screen* provides an interpretation of art as a form of occupation that serves as an ideological deflection within capitalism and may even profit concretely from labour stripped of its rights. This is how she considers that art can exploit political subjects such as feminism, gender equality and freedom in a critical manner (Steyerl 2012). However, in the context of feminist art, to consider representation and being on the side of minorities (or those who lost their basic rights), is also a responsibility of curatorial action, albeit that this responsibility neither guarantees the quality of action, nor even a clear political conclusion from it, but may only represent a will to change and offer an alternative to existing conditions.

Juli Carson's essay "Curating as a Verb: 100 Years of Nation-states" considers Arendtian philosophy as an applicable method in arts and says that according to Arendt, curating is a kind of thinking. "Each exhibition dealt with a present moment haunted by a past as a model of curatorial thinking. Moreover, each exhibition waged a secondary operation, a theoretical means of action, to achieve this thinking" (Carson 2020, p. 91). Carson starts with a brief account of Arendt, including her idea that there are no dangerous thoughts; thinking itself is dangerous (Baehr 2012). Carson describes how some philosophers, politicians, artists, writers, curators and others who use the act of curating to provide a thinking process and activate the distance between possible and impossible. An example of this might be the 11th Berlin Biennale.

Marion Von Osten, a feminist curator and scholar, considered Arendt's accounts in her analyses as a part of a broader strand of *labour* in "vita activa". In that respect, making or producing—in Hannah Arendt's sense of *Herstellen*—can only happen after the processes of dialogue, cognition, sketching, and study relates to the work made by the entire body concerning others (Von Osten 2018). In other words, the process of thinking and evaluating are the prerequisite to any action and run parallel to the freedom to act. The sum of these interpretations reveals a set of points about the nature of curatorial praxis: the need for a collective approach, the interpretative necessity to activate the distance between possible and impossible in a domain of freedom and production in relationship to parallel spaces, namely the work itself, the idea, the area, and the spectator.

However, the Arendtian perception of art is severely limited when it comes to discussing individual desire in artwork as she does not see how artworks are able to gather thoughts together in a plural way. For Arendt—following Kant—only aesthetic matters and aesthetics—and not the judgment of taste—belong to the realm of political and this does not have a connection with the arts (Arendt 1985). However, the opposite of Arendt's restricted perception about art is applicable to thinking about curatorial action, since as mentioned before, the curatorial is a way of thinking collectively with the aim of creating future change in society and setting up future political conditions by the means of art.

There are many critics of Arendt's understanding of the relationship between plurality, aesthetics, and political action. Jean-Luc Nancy believes Arendt does not define a specific functionality in public action, which might have broader influence on institutions. He criticizes Arendt's notion of the political act and indicates being with others is not a compulsory feature for producing political action, and instead, emphasizes how a responsibility to society is the lost piece of Arendt's puzzle of the political act (Nancy 2017). It would be beneficial to declare that here I have tried to emphasize the importance of collective political action as a sign of aesthetics in arts and as a feature in curatorial and vital element in the feminists' actions, to examine the comparison with other features, that Nancy determines as responsibility toward the public and ethics. Seyla Benhabib, by contrast, looks at the political action of Arendt in a practical way and emphasizes action as important in creating a place to negotiate political aims for the multitude (Benhabib 2018), which she sees as central to the notion of praxis at the heart of the desire for action.

As Susan Archer Mann writes, "Feminism is not simply a body of thought: it is a politics directed toward social change" (Archer Mann 2012). Accordingly, to connect action

and aesthetics within curatorial feminists' thoughts, Arendt's theory of action is beneficial. The fundamental features here are that it links public space to collective political action from one side and the curatorial action to the other parts of vita activa, including labour. As discussed earlier, curatorial action belongs to the present and future and acts in a collective manner in the presence of freedom, in the same way that feminism describes. Following that, since social and political action in a collective version is a pure notion of aesthetics in Arendt's theory, the feminists' turn in curatorial praxis will be an aesthetic phenomenon, too, due to common aims. This is why I am arguing that the closeness of feminist turn in the curatorial and Arendt's theory can be narrowed down to the meaning of action and, following this, that the existence of an aesthetic represents the desire for social and political change.

## 4. Four Case Studies

In the following paragraphs are a few examples of artworks and exhibitions whose subject is migration to further this discussion of the value of Arendt's theory of action when linked to aesthetics and the desire for sociopolitical change, with a focus on women's experiences.

Mieke Bal's video installation, Nothing is Missing (2006–2010), combines 17 videos shown on multiple screens in permanent loops, each 25–35 min long. Bal is both artist and curator of this work. Every video is devoted to a mother who talks about her migrated child who has left the motherland searching for a better life. Those women were filmed and witnessed by a close member of their family. This method of filming kept the conversation fluid. It enabled them to recount memories of their children comfortably and explain the reasons behind their move, their personal feelings regarding the separation, their perceptions about their children's new home, and what they know about the lives of their migrated children. These women sat in their homes wearing everyday clothes and spoke in their first language.

Mieke Bal chose to present videos on old TVs, emphasizing their story's nostalgic element. Spectators could see a woman on each TV while facing the camera and talking. The videos were unedited, and the sound was indistinct, as spectators could hear only a weak voice recording of the person in conversation with the woman. Even as they told their stories, the spectators fell silent, as if they were listening to the others' stories playing on the other TVs. English subtitles helped the audience to grasp these narratives. The ambient sound of various languages spoken simultaneously in the room created a deliberate anarchic situation. The installation enacted the tension between a global experience and an intimate one, given that Bal had created a domestic ambience within a public space (Bal 2015). As a result, *Nothing is Missing* has been presented in galleries, museums and many institutes around the world.

In *Nothing is Missing*, Bal collected videos from women interviewees to document pure memories with the minimum of her interference in leading the questions asked of the interviewees. Bal said of the project, "I am implicitly alluding to exploring social meaning" (Bal and Marx-Macdonald 2002). The entire project is an object for critical analysis of women in a particular situation. These videos documented the forgotten feelings of mothers who have experienced loss, disappointment, hope, failure, success, and challenges with their children's migration. They draw distance between the present moment of enunciation and the moment of the child's memory (Bal and Hernandez-Navarro 2008). This is where individual memories come together—without trivializing individual experiences or the uniqueness of their experiences as collected memories—and each finds their place in the bigger picture of women's history. Women's specific experiences and the history of society are overlapped with each other. Each woman emphasizes different features of their memory and their past events. The importance of this project is when the history is narrated by women from their personal experiences and memories. This approach stands against the normalized method of social historians, who try to use male-dominant social group categories to organize their perception of what has happened in the past.

*Nothing is Missing* is directly about women and, more specifically, mothers who suffered from separation from their children under the influence of sociopolitical conditions surrounding forced migration. Forced migration refers to people who are forcibly induced to move from their homeland and migrate to a new place to flee to escape conflict or persecution or have been trafficked. "The definition also encompasses situations of enforced immobility, for example, when displaced people are confined to refugee camps and detention centres. Forced displacement may occur within or across the borders of the nation-state. The effect of the force causing the migratory movement is crucial and distinguishes forced migrants—who may be termed 'refugee,' 'trafficked person,' 'stateless person,' 'asylum seeker,' or 'internally displaced persons' (IDPs)—from other migrants such as economic migrants" (Stankovic et al. 2021).

In this project, the act of Mieke Bal as the creator of the work is more curatorial than artistic since it is based on the process of its subjects focusing on the future rather than a single statement on the subject that looks backwards. The accuracy of the individual narratives is not a matter of further investigation by the artistic team because the plurality of women's narrations is the evidence for its truth that creates the work layer by layer. In other words, Bal's curatorial strategy produced a version of the truth about women that was not visible before her attempt. Yet, none of those mothers was representative of a featured community or had been selected to represent variations in class, country or profession; however, they were unified in being mothers and experiencing the same social event: the migration of a child abroad. Accordingly, I am emphasizing that Mieke Bal's approach is accumulating the cultural memory of a group of mothers rather than stories about migration as it has been written in male-dominant social history in order to understand the past differently.

Bal defined her method of showing the women's narratives as a plurality: "It cannot be a 'cinema of me' the first-person-singular documentary, but a heterogeneous first-person plural, where tasks are divided, but a collaboration replaces objectification" (Juhasz and Lebow 2020). The heterogeneous first-person plural is a good explanation to define the gesture of what has been achieved in *Nothing is Missing*. Each mother reflects on her perception of the past, present and the days to come; what she remembers; what she understood; and, in a nutshell, her comprehension of her child's departure story. Bal uses a technique in curatorial approaches that could be construed as revisionism. A revisionist approach rediscovers what the canon conceals and suppresses knowledge about a subject's perspective (Reilly and Lippard 2018). The revisionist method directly works for the present with some hopes for the future, which can be referred to the Arendt's action theory and the subjectivity of the future. In a lecture, "Thinking in Film", Mieke Bal used a sentence to define her approach, which can work as a sign of her method. Bal asserted that it is more about being "with it"—the subject—rather than "on it" to use more of the availability, which occurs when domains overlap (Bal 2015). Bal's revisionism in curating the exhibition gathered the diverse reflections of several women who experienced separation from their children. There was no attempt to aestheticize, fictionalise or manipulate the video content; however, the plural approach toward the number of narratives getting together with a common topic in the way that Arendt indicated a political action led the work toward the aesthetics in the Arendtian school of thought that was mentioned before.

The videos are reliable evidence of comprehensive curatorial action that discusses a sociopolitical event directly influencing women who were the hidden side of the migration story. It questions the adequacy of a generally accepted conceptual stance and focuses on the conceptual gaps to highlight discrepancies. The discrepancy Bal used to address the area she had concentrated on is definable as the unknown spaces among fragmented and inconsequential women's narratives.

Another example that addresses the discrimination against women's life is a project by Bahia Shehab called *20 Minarets from the Arab World–Call to Prayer*. This project is about ideological discrimination that has always influenced women who live in Islamic nation-states. Bahia Shehab's project picks one of the numerous inequalities in the ideological

states. *20 Minarets from the Arab World–Call to Prayer* (2014) is about Adhan or Azan, a Muslim chant to call everyone for prayers during the day. In this project, Shehab exposed women's existence as a part of society to have the right to be Adhan singers in the history of more than 1400 years of Islam. From the beginning of Islam's invention, only men were allowed to sing the chant to call for prayer. Hence, singing in public became entirely forbidden for women in many Islamic nation-states. Bahia Shehab raised the same Islamic chant with a Mezzo-Soprano woman opera voice in the exhibition and on her website, free to download[2] and ready to use in public. The project had an exhibition part, too, where Bahia painted the 20 important minarets in a round structure in the middle of the exhibition space in Louisiana Museum of Modern Art, Humlebæk, Denmark on 2014. The presentation consists of wall painting and audio art installation. She also included the minaret of the Great Mosque of Aleppo in Syria, but it appears to be in ruins to represent the cultural disaster that struck in 2013 when the minaret was bombed. In this project, Bahia was concerned with how the Arab cultural heritage within their nation-states was physically destroyed on the one hand. On the other hand, she showcased how it was intellectually attacked by Western nations and labelled as backward and terrorist (Shehab 2014).

A similar collective action recently took place in one of the music departments of the Art Universities of Tehran, the capital of Iran, where female students under the Islamic regime are not allowed to sing in public. Following the "Woman, Life, Freedom" movement that started on September 2022, university students, without any gender divisions, performed a famous revolutionary chant in support of the women leading movement in the main building of the campus—where it has been forbidden for any performances by women for many years—and invited spectators to join the female singers handing out a piece of paper with the lyrics. The name of the artist needs to remain private to protect their safety.

The exhibition *Mirrors of a Place, Views Gender and Immigration* (2015), curated by Tal Dekel, is an example of what happens in art projects that try to legitimize a situation differently when the concept of particular women's rights as an entirely political subject. This exhibition was a project about women who have migrated to Israel. The show occurred following the publication of *Women and Immigration, Art and Gender in a Transnational Age* (Dekel 2015). The curated project was presented in Tel Aviv and contained eleven women artists from diverse nationalities: the Philippines, Argentina, France, Iran, Ethiopia, and the former Soviet Union. The artworks shown were mainly videos and photographs. The exhibition's material, such as the catalogue and newsletter, was only published in Hebrew. Dekel's feminist perspective encouraged her to create an agency for women whose voices were silenced because of their gender and their status as immigrants without rights or any horizon for gaining citizenship in Israel because they are not Jewish or Jewish descendants. After 1948, Israel's policy implicitly changed to a nation-state based on the division of wanted and unwanted groups depending on their religions. It is essential to mention that it is true that Zionism never represented itself as a Jewish liberation movement but described itself as a movement for colonial settlement in the East (Lee 2000). "That famous slogan, *A land without people for a people without land*, has proven to be a performative rather than a descriptive statement, spelling the gradual ejection of a people whose persistent presence has been a perpetual obstacle to the completion of the Zionist project" (Lloyd 2012). This words emphasis the ideological plan for the Israeli regime that tried to gradually eliminate native Palestinians and other religious interests or assign them second-class citizenship in favour of Jewish settlers.

In the exhibition project, Dekel selected artworks for the exhibition from eleven migrant women artists. One of the artists was Luciana Kaplun, a woman who was a migrant from South America to Israel. She presented her 15-min video, *Mucamas* on 2011, which explores non-Jewish immigrant women residents' daily lives in Israel. The artist allowed these women to talk or do whatever they wanted. The only important thing was to show the real experiences of their lives, even in a fictional way (Dekel 2015). The artist effectively reveals the implicit connections between citizenship and the sense of

belonging while emphasizing the different statuses of migrant women who came to the same nation-state. Like Kaplun's video, other works in *Mirrors of a Place* showed issues of women's identity, rights of migrants and the actual situation of those with migration status and without a specific religion. Sometimes, subjects tended to show only women or their perspectives; but many also foregrounded issues of religious observance, ethnicity, nationality or class. Although Dekel's exhibition consists of a few art pieces, Arendt's concept of curatorial action does not appear in this project, because the result is a collection of artworks rather than a collective curatorial project. Since the curatorial here does not refer to the professional work of curators alone but addresses the analytical process of feminist curating as an approach to action. Therefore, according to my analyses, an artwork project conducted by an artist has the feasibility of being a curatorial project and a collective project conducted by curators has the possibility of ending up as an artwork. One of the main reasons for this criticism is the lack of communication about the conception of the idea for the exhibition from the group of artists exhibited. Another problematic aspect of Dekel's project is the lack of a desire for change. The exhibited works remained within the boundaries of storytelling and could as a result only address the past.

Contrary to the Dekel project, Tanja Ostojic's process-based work has all the features defined above about the feminist concept of curatorial action. *Misplaced Women* is an intervention that challenged the limitations of collective action. The project's website[3] became active in 2008. It is a web project that combined multimedia, installation, workshops and communication platforms. The body of the project was created in 2008 and is still ongoing. *Misplaced Women* is an online platform available to all women interested in participating in one or more project sections. Every woman with internet access can become involved in one or more parts. The project focuses on some of the daily activities that thematise displacement, known to migrants, refugees and itinerants travelling the world to earn their living. "Those performances deal with migration issues, gender democracy, feminism, gentrification, inclusion, power relations and vulnerability, particularly concerning the female and transgender bodies".[4]

The curatorial praxis is not unified or classified; however, the grounding for this method is that its inquiry is interdisciplinary. An art exhibition or in the case of Ostojic's project, *Missing Women*, where the artistic idea is a medium of presenting the thoughts and when it becomes collective, is defined by action. As Arendt indicates, there are no dangerous thoughts; thinking itself is dangerous (Baehr 2012); therefore, considering this proposition, the ontology of the relationship between components is crucial in order to define action within this project as both a curatorial one and a social practice. The informative blog and workshops offer community practice and relevant skills to feminist emancipatory methodologies as noncommercial services to women in an artistic approach that Ostojic selected. The target audience varied from women artists to the invisible communities such as migrants, asylum-seekers, retired individuals and unemployed members of society. Going back to Arendt's theory of action and curatorial features discussed before, curatorial action is a form of potential change. The desired changes were shaped in this project by the flexible platform where the practice is tied to that particular Agora—in Arendt's words—and by considering relevant regional factors and essential subjects. Following that idea, *Misplaced Women* suggests a functional new version of an Agora that was fluid but also had minimum limitations for its members. On the other hand, it did provide a form of criticism in its conversations and dialogues about the existing situation and looked for an alternative to create change by utilizing women's knowledge and experiences.

## 5. Conclusions

Regarding distinguishing artistic from curatorial projects in relation to feminist action; some of the case studies have explored the artist's personal experiences of in relation to the gender subject (Bahia Shehab and Tal Dekel's project). In others, the project explores a specific subject's experience or directly follows a strand of feminists' theory from social science and politics (Bal's and Ostojic's projects). In some cases, the artists led and were

effectively the curators of their own exhibitions, such as Mieke Bal's work. There are other examples that try to make activist / political change through the project—in terms of perceptions, representation or behaviour—in relation to feminists' action, such as Bahia Shehab and Ostojic's works. The consequences of each step in the process of their projects reveals the next part, and that is what leads the project forward in both curatorial and artistic decision-making as well as its desire to invent an action.

I have shown how the collective praxis of "the curatorial" and art projects/exhibitions toward women subjects validate an aesthetics in collective curatorial action and tried to evaluate the outcome of Arendt's conception of action and artwork in each case study. Feminist theory indicates the importance of women's actions in order to break boundaries, recognize the various types of patriarchy and male dominations, and become political actors. In this regard, art can provide and develop a platform to examine diverse social and political actions in the form of praxis that directs the participants and spectators toward new contemporary definitions of aesthetic activities. One of the fundamental notions is that "the curatorial" here does not only refer to the professional work of curators but addresses the analytical process of "the curatorial" as an approach. Collective thoughts are the starting point of a unique type of social praxis and is an important portion of feminists and curatorial action that in Arendtian school of thought, is a vital feature of aesthetics. This collective political desire has been constitutive to the emergence of feminist art. Curatorial thoughts have already been profoundly entangled with political and social questions and, consequently, involve action and aesthetics. Following the arguments and theories about curatorial action in the sociopolitical subject and, more specifically, art about women's condition in this research, my argument aims to articulate debates identifying curatorial actions in curated projects looking for collective action. Hannah Arendt's definition of action, and her commentators make these distinctions clearer.

Feminist curatorial projects by both artists and curators can go beyond the boundaries of the time and place of the exhibition and step into the realm of discursive desire for change, especially by thinking in terms of action in both political and art realms. A collective experimental tendency in the curatorial provides a safe ground to evaluate these different elements in emerging feminist artworks and exhibitions and opens new ways to experience aesthetics in real life through social action.

**Funding:** This research received no funding.

**Data Availability Statement:** Not applicable.

**Conflicts of Interest:** The authors declare no conflict of interest.

## Notes

1   This article is a part of a larger study from the author's PhD thesis titled "A curatorial study of migration in arts" (2018–2022), Middlesex University, London, UK.
2   See: https://fineacts.co/bahia-shehab (accessed on 6 September 2021).
3   See: www.misplacedwomen.wordpress.com (accessed on 6 September 2021).
4   See: https://misplacedwomen.wordpress.com/about/ (accessed on 6 September 2021).

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
