# Peer review of "Hannah Arendt’s Action Theory, Aesthetics and Feminist Curatorial Praxis"

_arts, 2022_

Round 1

Reviewer 1 Report

The article identifies an important lacuna in the political theory of curating by turning to Hannah Arendt’s theory of action and bringing it to a specifically feminist curatorial praxis. This is of interest in relation to the historical tensions around ‘woman’ as ‘political subject’ and realities of women defined through conditions of citizenship. Future work around this could explore in more detail how the concept intersectionality can not only expand Arendt's theory of action, but also bring conflict and tensions into her theorization. 

Author Response

Dear all,

Many thanks for your suggestions. I have applied all of them and would like to thank you for helping me improve the quality of this manuscript.
Please find below the reply to your comments:

-  Rewriting lines 355-358 is key but more explanation (and maybe a quote from Arendt) would also be productive:
Revised (New 383-389):  The curatorial praxis is not unified or classified; however, the method and sharing ground with other disciplines bring its inquiry to the ground. An art exhibition or in the case of Ostojic’s project, the artistic idea is a medium of presenting the thoughts and when it becomes collective, steps in action domain. Arendt indicates, there are no dangerous thoughts; thinking itself is dangerous (Arendt 2012), therefore, considering this proposition, the ontology of the relationship between components is crucial in order to define an action as a curatorial one or as a social practice.  The informative blog and the workshops offer community practice and relevant skills to feminist emancipatory methodologies as non-commercial services to women in an artistic approach that Ostojic selected.

- 56-58: (New: 55-61): Added: Equally, the potential for dialogue, exchanging ideas, and movements between different times, sites, and perspectives as provided by feminist theory is central to theorising, and exercising feminist approaches to the curatorial. Amended: Curatorial praxis can become activated in the social and cultural gaps between existing models of knowledge, and because it is linked to the domain of thinking, it is able to create a space for a new language in arts to examine theories in real life. 

 - 64: deleted: The trigger for the existence and expansion of this creativity is the long history of living under oppressive forces and gender injustice worldwide.

- 114-118: (New 119-120): changed to: Accordingly, the two central notions of action are freedom and plurality. Action can only be understood in the communal version of society and by grasping and analysing this notion of action, any concept of action by one individual becomes impossible. In this regard, action can be recognized as similar to language in that both become activated only in a collective community. 

- 130-132: (New 141-144): Clarified:  'Arendt’saction theory relies on the present but it is also a process that directlyinfluences the future. Unlike other philosophers, Arendt’s philosophy hasrarely been central to curators’ work and so few directly refer to herphilosophies since her philosophy considered more relevant to the social andpolitical science than art.'  

- 161-162 (New 136-140): 'Here the matter is how and when artists create complex projects that require a curator to follow their lead and how and when curators bring artists together to create artistic ensembles of meaning. This is why the curatorial doesn’t refer to being a member of a profession, but to a way of artistic thinking and behaving with the aim of produc-ing/representing collective action.' Replaced with: 'Curating, therefore, can be interpreted as a self-generating process that prompts a cluster of new thinking for the spectator.'

- 165-166: Added: An example of this might be

 - 173-180: (New 177-198): Added: However, the Arendtian perception of art is severely limited when it comes to discussing individual desire in the artwork as she doesn’t see how artworks are able to gather thoughts together in a plural way. For Arendt – following Kant - only aesthetic matters and aesthetics - and not the judgment of taste - belong to the realm of political and this doesn’t have a connection with the arts (Arendt 1985). However, the opposite of Arendt’s restricted perception about art is applicable to thinking about curatorial ac-tion, since as mentioned before, the curatorial is a way of thinking collectively with the aim of creating future change in society and setting up future political conditions by the means of art. 
Amended: There are many critics of Arendt’s understanding of the relationship between plurality, aesthetics, and political action. Jean- Luc Nancy believes Arendt does not define a specific functionality in public action which might have broader influence on institutions. He criticizes Arendt’s notion of the political act and indicates being with others is not a compulsory feature for producing political action, and instead, emphasizes how a responsibility to the society is the lost piece of Arendt’s puzzle of the political act (Nancy 2017). It would be beneficial to declare that here I have tried to emphasize the importance of collective political action as a sign of aesthetics in arts and as a feature in curatorial and vital element in the feminists’ actions, to examine the comparison with other features, that Nancy determines as responsibility toward the public and ethics. Seyla Benhabib, by contrast, looks at the political action of Arendt in a practical way and emphasizes action as important in creating a place to negotiate political aims for the multitude (Benhabib 2018) which she sees as central to the notion of praxis at the heart of the desire for action. 

- 187-188: Deleted: Consequently, the collective action of Arendt and the desire for a sociopolitical change become a part of the curatorial.

- 218-219:  Added: Bal is both artist and curator of this work. 

- 228-231: (New 248-253): Changed to: Women's specific experiences and the history of society are overlapped with each other. Each woman emphasizes different features of their memory and their past events. The importance of this project is when the history is narrated by women from their personal experiences and memories. This approach stands against the normalized method of social historians who try to use male-dominant social group categories to organize their perception of what has happened in the past.

- 234: (New 256-264): surrounding forced migration. Forced migration refers to people who are forcibly induced to move from their homeland and migrate to a new place to flee to escape conflict or persecution or have been trafficked. ‘The definition also encompasses situations of enforced immobility, for example, when displaced people are confined to refugee camps and detention centres. Forced displacement may occur within or across the borders of the nation-state….the effect of the force causing the migratory movement is crucial and distinguishes forced migrants—who may be termed “refugee,” “trafficked person,” “stateless person,” “asylum seeker,” or “internally displaced persons” (IDPs)—from other migrants such as economic migrants’’. (Stankovic, 2021). 

 - 244-245: (New 274-277): Added: Accordingly, I am emphasizing that Mieke Bal’s approach is accumulating the cultural memory of a group of mothers rather than stories about migration as it has been written in male-dominant social history in order to understand the past differently. 

- 264-7 (New 294-297): There was no attempt to aestheticise, fictionalise or manipulate the video content; however, the plural approach toward the number of narratives getting together with a common topic in the way that Arendtindicated a political action led the work toward the aesthetics in theArendtian school of thought that has been mentioned before. 

- 273-5, Deleted: Put differently, focusing on the discrepancy in the heart of the culture and the will of change - here to be heard- reveal the links between aesthetics, feminist idea and curatorial action.

- 296-303: (New 324-332): Revised: A similar collective action recently took place in one of the music departments of the Art Universities of Tehran, the capital of Iran, where female students under the Islamic regime are not allowed to sing in public. Following the “Woman, Life, Freedom” movement that started on September 2022, university students, without any gender divisions, performed a famous revolutionary chant in support of the women-leading movement in the main building of the campus - where it has been forbidden for any performances by women for many years - and invited spectators to join the female singers handing out a piece of paper with the lyrics. The name of the artist needs to remain private to protect their safety. 

- 339: (New 367-373): Added: Dekel’s exhibition consists of a few art pieces; Arendt’s concept of curatorial action doesn’t appear in this project, because the result is a collection of artworks rather than a collective curatorial project. Since the curatorial here doesn’t alone refer to the professional work of curators alone but addresses the analytical process of feminist curating as an approach to action. Therefore, according to my analyses, an artwork project conducted by an artist has the feasibility of being a curatorial project and a collective project conducted by curators has the possibility of ending up as an artwork. 

- 355-358 : (New 390-396): Added: The curatorial praxis is not unified or classified; however, the grounding for this method is that its inquiry is interdisciplinary. An art exhibition, or in the case of Ostojic’s project, Missing Women, where the artistic idea is a medium of presenting the thoughts and when it becomes collective, is defined by action. As Arendt indicates, there are no dangerous thoughts; thinking itself is dangerous (Arendt 2012); therefore, considering this proposition, the ontology of the relationship between components is crucial in order to define action within this project as both a curatorial one and a social practice. 

- In conclusion, this paragraph is added at the beginning: Regardingdistinguishing artistic from curatorial projects in relation to feminist action, some of the case studies have explored the artist’s personal experiences in relation to the gender subject (Bahia Shehab and Tal Dekel’s project). In others, the project explores a specific subject’s experience or directly follows a strand of feminists’ theory from social science and politics (Bal andOstojic projects). In some cases, the artists led and were effectively the curator of their own exhibition, such as Mieke Bal’s work. There are other example that tries to make activist / political change through the project – in terms of perceptions, representation or behavior -  in relation to feminists action, such as Bahia Shehab and Ostojic’s works. The consequences of each step in the process of their projects reveal the next part, and that is what leads the project forward in both curatorial and artistic decision-making as well as its desire to invent an action.
revised: One of the fundamental notions is that ‘thecuratorial’ here doesn’t alone refer to the professional work of curators but addresses the analytical process of ‘the curatorial’ as an approach. Collective thoughts are the starting point of a unique type of social praxis and are an important portion of feminists and curatorial action that, in the Arendtian school of thought, is a vital feature of aesthetics. This collective political desire has been constitutive to the emergence of feminist art. 

Reviewer 2 Report

This paper begins with a compelling premise but it is a challenging one to explicate conceptually and the author has not fully succeeded. As they admit in the conclusion, this argument works better in the theoretical domain than in the real world of curating. More problematic here is the argumentation, which is not sufficient to demonstrate the topic with clarity, or to apply it successfully in the case studies. Trying to use Arendt's theories to make an argument about feminist curating is an uphill battle and the author has not been able to successfully play this idea through in the piece as it is. I do not believe it would be impossible to make this case, but it must be more effective than what was submitted. I will devote the rest of these comments to pointing out areas where I think the argument could be improved and introducing some questions to specific points that have been made.

The fundamental terms of this argument (Arendt's notion of action, the curatorial, and feminism) are all a bit nebulous in this account. I think it would help to quote Arendt since the author is drawing their foundational concepts from her work and the text has not adequately expressed the complexity of her thoughts on action and aesthetics, as they might be applied to the curatorial. Further, the author is working with unreconstructed notions of both feminism and curating. The inequalities lurking in feminist claims have been a target among feminists of color since at least the late 60's and, in more recent years, the concept of intersectionality has done a lot to transform and diversify thinking on feminism. The whole notion of gender and sex has been put up for grabs more recently, leading to a wide array of responses among feminist scholars. But none of that is even referenced in the author's discussion of feminist curatorial practice. Curating has less of a history but there have been many books and articles on the curatorial turn in the 21st century. Many of the ideas expressed in this piece have been challenged in this literature but, once again, there are no references to these debates. As someone who worked as a museum curator for 10 years, what is said about curating here has little relevance to the work I was engaged in. The qualification placed in the conclusion would be better used in the introduction. It is an idea of curating that is at stake here, and an idea of action from Arendt. These are theories and, like theoretical work on human rights, they have little relation to the way the ideas are put into practice in the world.

This is why the case studies fall flat--when one attempts to apply these ideas to exhibitions that have happened, the author should be moderating and adjusting the theories based on how the practice operates, but instead the particulars of these exhibitions are glossed over and the theories are not elaborated as a result. There is a fundamental confusion about the difference between curating and social practice (in the artistic sense). A curator tells a story but uses artists who have made works to elaborate that story. An artist collects artefacts and evidence from the world and presents them in a compelling way. Authorship exists in both contexts but curators are generally thought to work with art works made by others. Thus describing Bal's project, it seems that the author has engineered this project, rather than drawing on the work of one artist or a group of them, which means she is not curating as conventionally understood. If the author wants to argue that this is curating, rather than social practice, that argument needs to be made here (and it would be helpful to clarify this more in the discussion of Arendt and the curatorial in the first section). In the Call to Prayer piece, here again it is an artist who is making not only a social practice work, but other visual elements to tell the story that sound like more conventional works of art. I am sure there was a curator of this show at the Louisiana museum but that is not even mentioned here; instead there is an implicit (not explained) assumption that this artist is curating. (I believe this is based on the author's interpretation of Arendt, but it is not sufficiently explained.) The third example actually does sound like curating but the exhibition is not well articulated and in the end the author makes the opposite claim. In the last example, the author makes a good attempt to bring Arendt, the theory of action, and the concept of equality at stake back into the argument. This could be clearer though, and it would be useful if the author could use this example to elaborate how the case studies both affirm and challenge the conceptual model they want to employ. Rewriting lines 355-358 is key but more explanation (and maybe a quote from Arendt) would also be productive.

Please clarify or elaborate claims made in 

lines 56-58, 64, 114-118, 130-132, 161-162, 173-180, 187-188, 228-231, 244-245, 264-7, 273-5, 296-303, 339, 355-8

234 -forced migration, as in refugees? If so, then the term migrant above is inaccurate

Author Response

(The authors gave the same response as above.)

Reviewer 3 Report

The draft has a solid theoretical background. However, I am unsure how we can talk about curatorial projects, feminist proxies, aesthetics, and women's experiences without any supporting visual data, original contribution and updated case studies. Mainly in part 4 (Case studies part), the core part of the research, the author(s) uses the example of four curatorial case studies without any visual materials. On the other hand, rather than collecting the other published, mostly outdated case studies, the case study part still needs the more direct and actual engagement of the author(s). It would be beneficial to use the author(s) curatorial projects or those in which the author(s) had direct engagements and involvements. Also, please support the case study part with the required visual materials and use updated projects and examples. 

Author Response

(The authors gave the same response as above.)

Round 2

Reviewer 2 Report

This new draft represents a major revision of the piece which clarifies a lot of the issues that were previously nebulous. The article has not been reinvented but it has been substantially rewritten and the author has made a number of passages that were confusing much clearer, leading to a far more satisfying analysis. The amazing thing is how quickly this was achieved.

I have one quibble, which is that the author makes a very different analysis of Bal's and Dekel's projects--one represents action and the other does not-- but they are both efforts to do a very similar thing: to give a voice to the marginalized women in the contemporary world (granted Dekel's project is focused only on Israel). I would be interested for the author to consider/explore the difference between these two initiatives/curatorial projects just a bit more because therein lie some very compelling analytical tools to parse feminist action in the domain of the curatorial. There is more explanation/analysis of Dekel's show that has been added and this is productive. But if Bal is rewriting social history from the perspective of mothers, why does the author think that Dekel is not also reconfiguring this model of history? How is commissioning artist/migrants to make their own works different than interviewing women for presentation at a gallery exhibition? Is the agency of the artist at odds with the agency of the curator? I suppose Arendt would think so but I think this is a key point to make in comparison of the two projects.

Author Response

Thank you for the comments